# MiR-92 Family Members Form a Cluster Required for Notochord Tubulogenesis in Urochordate *Ciona savignyi*

**DOI:** 10.3390/genes12030406

**Published:** 2021-03-12

**Authors:** Libo Yang, Xiaoming Zhang, Chengzhang Liu, Jin Zhang, Bo Dong

**Affiliations:** 1Sars-Fang Centre, MoE Key Laboratory of Marine Genetics and Breeding, College of Marine Life Sciences, Ocean University of China, Qingdao 266003, China; ylb8010875@163.com (L.Y.); zhangxiaoming@ouc.edu.cn (X.Z.); zhangjincoolcool@163.com (J.Z.); 2Laboratory for Marine Biology and Biotechnology, Qingdao National Laboratory for Marine Science and Technology, Qingdao 266237, China; 3Key Laboratory of Experimental Marine Biology, Institute of Oceanology, Chinese Academy of Sciences, Qingdao 266071, China; liu@qdio.ac.cn; 4Institute of Evolution and Marine Biodiversity, Ocean University of China, Qingdao 266003, China

**Keywords:** miRNA cluster, miR-92, *Ciona*, notochord, lumen formation

## Abstract

MicroRNAs are frequently clustered in the genome and polycistronically transcribed, regulating targeted genes in diverse signaling pathways. The miR-17-92 cluster is a typical miRNA cluster, playing crucial roles in the organogenesis and homeostasis of physiological processes in vertebrates. Here, we identified three miRNAs (csa-miR-92a, csa-miR-92b, and csa-miR-92c) that belonged to the miR-92 family and formed a miRNA cluster in the genome of a urochordate marine ascidian *Ciona savignyi*. Except for miR-92a and miR-92b, other homologs of the vertebrate miR-17-92 cluster members could not be identified in the *Ciona* genome. We further found that the mature sequences of urochordate miR-92 family members were highly conserved compared with the vertebrate species. The expression pattern revealed that three miR-92 family members had consistent expression levels in adult tissues and were predominantly expressed in heart and muscle tissue. We further showed that, at the embryonic and larval stages, csa-miR-92c was expressed in the notochord of embryos during 18–31 h post fertilization (hpf) by in situ hybridization. Knockout of csa-miR-92c resulted in the disorganization of notochord cells and the block of lumen coalescence in the notochord. Fibroblast growth factor (FGF), mitogen-activated protein kinase (MAPK), and wingless/integrated (Wnt)/planar cell polarity (PCP) signaling pathways might be involved in the regulatory processes, since a large number of core genes of these pathways were the predicted target genes of the miR-92 family. Taken together, we identified a miR-92 cluster in urochordate *Ciona* and revealed the expression patterns and the regulatory roles of its members in organogenesis. Our results provide expression and phylogenetic data on the understanding of the miR-92 miRNA cluster’s function during evolution.

## 1. Introduction

MicroRNAs (miRNAs), a class of small, endogenous, noncoding RNAs [1,2], play an essential role in multiple biological processes by regulating gene expression at the posttranscriptional level [3,4,5]. Approximately 37% of human miRNA genes localize as clusters in the human genome [6]. MiRNAs in a cluster are transcribed together with physically adjacent miRNAs and usually have similar functions or participate in regulating the same signaling pathway [7].

The miR-17-92 cluster is one of the best-studied miRNA clusters and was the first recognized as an oncogenic gene in human B-cell lymphomas [8]. This cluster localizes at chromosome 13q31 and comprises six mature miRNAs, including mir-17, mir-18a, mir-19a, mir-20a, mir-19b-1, and mir-92a-1, in humans [9,10,11]. The organization and sequences of the miR-17-92 cluster are highly conserved among vertebrates. The gene duplication and deletion events resulted in two paralogs of mir-17-92 cluster: mir-106b-25 and mir-106a-363 [12] during early vertebrate evolution. The two homologous clusters encode 15 different miRNAs (mir-106b, mir-93, mir-25, mir-106a, mir-18b, mir-20b, mir-19b-2, mir-92a-2, and mir-363), and these miRNAs share similar highly conserved gene sequences with the miR-17-92 cluster [13]. Recent studies have uncovered that the miR-17-92 cluster plays various roles in vertebrate organ development [14,15]. For example, the miR-17-92 cluster post-transcriptionally regulated the expression of Friend of Gata-2 (FOG-2) and inhibited mouse cardiac development [16]. The overexpression of miR-17-92 promoted cardiomyocyte proliferation in postnatal and adult hearts [17]. Besides, the miR-17-92 cluster was critical for muscle development. The mir-17 and mir-19 cooperatively promoted the repair of mouse tibialis anterior muscles [18]. Mir-92a regulated the rho-associated coiled-coil-forming kinase/myosin light chain kinase (ROCK/MLCK) signaling pathway to promote vascular smooth muscle cell proliferation and migration [19].

In invertebrates, the miR-17-92 family members seem to play similar regulatory roles in organogenesis as they do in vertebrates, but they do not form a cluster in the genome. During *Drosophila* development, the deletion of mir-92b induced an abnormally high expression of myocyte enhancer factor 2 (*Mef2*), leading to muscle defects [20]. miR-92a and miR-92b directly targeted jing interacting regulatory protein 1 (*jigr1*) to maintain neuroblast self-renewal through inhibiting premature differentiation [21]. In *Caenorhabditis elegans*, miR-235, a sole ortholog of mammalian miR-92 from the miR-17-92 cluster, played a vital role in coupling blast-cell behaviors to the nutritional state [22]. In oyster *Crassostrea gigas*, mir-92d regulated the expression of tumor necrosis factor during the inflammatory response [23].

Urochordate *Ciona*, the vertebrates’ closest relatives, possesses a simple chordate body plan and a small genome size. Hundreds of known and novel miRNAs have been identified in *C. savignyi* [24], *C. robusta* (former name: *C. intestinalis* type A) [25,26,27], *Halocynthia roretzi* [28], and *Oikopleura dioica* [29]. The notochord of *Ciona* larvae is a straight lumen-filled tube [30]. Previous studies showed that miRNAs were involved in the regulation of notochord development. Some miRNAs regulated the process of lumen formation and expansion, which happened in *Ciona* notochord development. For example, MiR-342 targeted the inhibitor of DNA-binding 4 (ID4) and DNA methyltransferase 1 (DNMT1) to regulate the lumen formation in mammary gland morphogenesis [31]; miR-26a mediated the vascular endothelial growth factor (VEGF) and promoted lumen formation via PI3K/AKT and MAPK/ERK pathways in rat angiogenesis [32]. However, the expression and function of the miR-92 cluster in notochord development remain largely unexplored. We previously built miRNA libraries to screen those miRNAs involved in regulating embryogenesis and larval metamorphosis in ascidian [24]. During the screening, the expression of the miR-92 family members were identified to be highly upregulated [24]. We hereby focused on this miRNA family and investigated the structure and functions of its members during development in ascidian. Here, we reported the identification of a miR-92 cluster in the *Ciona* genome and its expression and roles in notochord development.

## 2. Materials and Methods

### 2.1. Animals, Tissues, and Embryos

Adults of *C. savignyi* were collected from the coast of Jiaozhou Bay, Qingdao, China. The animals were maintained in a ventilating tank with continuous light in the laboratory.

Adults were dissected to collect different tissues, including gonad, intestinal, heart, blood cells, muscle, egg, and stomach. The collected tissues were immediately frozen in liquid nitrogen for RNA extraction.

*Ciona* eggs were collected from the dissected adults and then mixed with sperm from other individuals in seawater for fertilization for five minutes. The embryos and larvae were cultured at 16 °C and then were collected at different stages for RNA extraction, in situ hybridization, and immunofluorescence staining.

### 2.2. Quantitative Real-Time PCR (qPCR)

Total RNA was extracted from different stages of embryos or frozen tissues using RNAiso plus (TAKARA, Japan). The reverse transcription was carried out using 1-μg total RNA from different stages of embryos or frozen tissues by the miRNA 1st Strand cDNA Synthesis kit (Vazyme, Nanjing, China). The distributions of miR-92s at different stages and in different tissues were analyzed by 2-step qPCR (Vazyme, Nanjing, China) using a Light Cycler 480 (Roche, Basel, Switzerland). The reaction condition was as follows: 95 °C (5 min) for initial denaturation and then followed by 40 cycles at 95 °C for 15 s and 60 °C for 1 min. U6 snRNA served as an internal normalization control. At least three templates were analyzed to guarantee the accuracy of the experimental results. All data were calculated by the 2^−ΔΔCt^ method, and statistical analyses were performed using paired Student’s *t*-tests. The graph of qPCR results was plotted using Origin software (www.originlab.com, accessed on 24 August 2019).

### 2.3. In Situ Hybridization

The DIG-labeled locked nucleic acid (LNA) probes were synthesized by JIE LI BIOLOGY. Ascidian embryos at 10, 18, 21, 24, 31, and 42 h postfertilization (hpf) were collected and fixed in 4% paraformaldehyde (PFA) overnight at 4 °C. The embryos were permeabilized by digesting with 12-μg/mL proteinase K at 37 °C for 15 min. Subsequently, in situ hybridization was performed at a temperature 20 °C below the probe melting temperature for 18 h using csa-miR-92c LNA probes in a hybridization solution. After hybridization, samples were washed in gradient saline–sodium citrate at the hybridization temperature. Hybridization signals were detected using alkaline phosphatase-conjugated digoxigenin antibody (Roche, Basel, Switzerland) at a 1:2000 dilution. Embryos were stained with the BCIP/NBT substrate system (Roche, Basel, Switzerland) and visualized under the microscope.

### 2.4. Constructions and Electroporation

Three target sequences of CRISPR/Cas9 against csa-miR-92c were designed by CRISPRdirect (http://crispr.dbcls.jp, accessed on 1 December 2019). The sequences of selected single guide RNAs (sgRNAs) are listed in Appendix A, and the sequence of control sgRNA was designed according to the previous study [33]. Based on the target sequences of CRISPR/Cas9, all sgRNAs were synthesized and cloned into the vector of Cr-U6>sgRNA(F+E) (Addgene number: 59986) for expressing the sgRNA, respectively. The CRISPR/Cas9 system included the PCR product of sgRNA and the plasmid of Cs-Brachyury(3k)>NLS::Cas9::NLS::P2A::mCherry. Electroporation was performed according to the method described previously, with some modifications [34]. Briefly, the PCR product of sgRNA (50 μL), the plasmid of Cs-Brachyury(3k)>NLS::Cas9::NLS::P2A::mCherry (30 μg), the embryos (300 μL), and the electroporation buffer (420 μL) were electroporated into fertilized eggs at the following conditions: the capacitance was 1500 μF, the voltage was 50 V, and the time constant was within the range of 15–20 ms. After electroporation, the eggs were transferred to a fresh seawater agarose dish, and the embryos were allowed to develop at 18 °C to the desired stages (16 hpf, 21 hpf, and 31 hpf) for confocal observation.

### 2.5. Validation of the Efficiency of the gRNAs

The PCR product of sgRNA and the plasmid of Cr-EF1α>NLS::Cas9::NLS and Cr-EF1α>NLS::Cas9::NLS::mCherry were co-electroporated into the fertilized eggs. After electroporation, embryos were cultured for about 12 h at 18 °C. The embryos with red positive signals were collected and lysed by a direct PCR lysis buffer at 60 °C for 10 min. PCR was performed from the lysate using Phanta^®^ Max Super-Fidelity DNA Polymerase (Vazyme, Nanjing, China) as follows: 55 °C annealing temperature for 15 s and 72 °C extension temperature for 30 s, 35 cycles. PCR products were purified by the GeneJET Gel Extraction Kit (Thermo Fisher, Waltham, Massachusetts, USA). Two hundred-nanogram PCR products in the knockout (KO) group and 200-ng control PCR products were diluted to 17 μL by ddH_2_O and mixed with 2-μL T7 reaction buffer (Vazyme, Nanjing, China) in a 200-μL PCR tube and incubated with 1-μL T7 endonuclease I (Vazyme, Nanjing, China) at 37 °C for 30 min. The products were detected on a 2% agarose gel at 100 V for 30 min, and the image of the gel was analyzed by ImageJ (National Institutes of Health). Cleavage efficiency was determined according to the previously published protocol [33]. The PCR product of mutant gDNA was ligated into the pEasy Blunt-3 vector (Transgen, Beijing, China) for Sanger sequencing (GENEWIZ, Suzhou, China).

### 2.6. Immunofluorescences

The fixed embryos were washed three times in PBST and then incubated with 1/100 dilution of phalloidin overnight at 4 °C. After removing the antibody solution, the samples were washed three times in PBST at room temperature. DAPI staining was performed in the dark for 30 min. The samples were imaged using a confocal laser scanning microscope.

### 2.7. Bioinformatics Analysis

The precursor and mature sequence of miR-92 from different species were extracted from the miRNA database miRbase22.1 (http://www.mirbase.org/, accessed on 14 August 2020). The Ensembl database (http://asia.ensembl.org/, accessed on 14 August 2020) was used to analyze the distribution of the miR-92 family or cluster members in the genome. The secondary structure for “csa-mir-92” was predicted using miR-Deep2 and miReap (https://sourceforge.net/projects/mireap/, accessed on 14 August 2020). For the evolutionary analysis, the multiple sequence alignment was performed using CLC Main Workbench 5. The phylogenetic tree was constructed using the Maximum-Likelihood (ML) method by the Molecular Evolutionary Genetic Analysis of MEGA X.

For target gene prediction, 3′ UTRs of *C. savignyi* mRNAs obtained from previous RNA-seq data [24] were used as potential target sequences. MiRNA-binding sites were predicted using miRanda and targetscan 7.1 (targetscan.org, accessed on 24, August, 2019). The underlying functions and vital pathways of the target gene candidates were analyzed by Gene Ontology (GO) functional annotation and Kyoto Encyclopedia of Genes and Genomes (KEGG) pathway analyses (www.kegg.jp/kegg/kegg1.html, accessed on 14 August 2020). The OmicShare tools were used to test the statistical enrichment of the target gene candidates in the GO and KEGG pathways. The calculated *p*-value went through False Discovery Rate (FDR) Correction, taking FDR ≤ 0.05 as a threshold.

### 2.8. Transcriptome Sequencing

Ascidian embryos and larvae were collected at 18, 21, and 42 hpf and were then immediately frozen in liquid nitrogen. Total RNA was extracted with RNAiso Reagent (Takara, Beijing, China). in total, 3 μg of total RNA per sample was used as the input material for the construction of small RNA libraries. Libraries were generated using NEBNext^®^ Multiplex Small RNA Library Prep Set for Illumina^®^ (NEB, Beijing, China) following the manufacturer’s instructions and were sequenced on an Illumina Hiseq 2500/2000 platform at Novogene, China.

## 3. Results

### 3.1. Characteristics of csa-miR-92 in C. savignyi

We firstly blasted the mature sequences of miR-92 from 12 species in the miRbase database, including *C. savignyi*, C. robusta, *Homo sapiens*, *Mus musculus*, *Gallus gallus*, *Drosophila pseudoobscura*, *Xenopus laevis*, *Danio rerio*, *Branchiostoma floridae*, *Saccoglossus kowalevskii*, *Capitella teleta*, and *Strongylocentrotus purpuratus* (Appendix A), and then made a multiple sequence alignment of synteny for both the paralogs and orthologs. Eventually, we identified three miR-92 family members, including csa-miR-92a, csa-miR-92b, and csa-miR-92c, in the *C. savignyi* genome. All these three miRNAs located in the intergenic region on reftig_1: 1335375-1336487 contig. Csa-miR-92b and csa-miR-92c were separated by 81 base pairs, while csa-miR-92c and csa-miR-92a were 769 base pairs apart in the genome, forming a cluster (Figure 1A).

Furthermore, the aligned data showed that the consensus sequence was the “UAUUGCACUUGUCCCGGCCUG-U” (Figure 1B), indicating that the mature miR-92 sequences were highly conserved, while its seed sequences were the same and likely shared the same targeting properties across species. The secondary structures analysis showed six, nine, and six local stem-loop structures in csa-mir-92a, csa-mir-92b, and csa-mir-92c, respectively (Figure 1C–E). The sequence alignment and secondary structure suggest that csa-mir-92a, csa-mir-92b, and csa-mir-92c belong to the miR-92 family. The physical location in the chromosome and the conserved structure characteristics indicate that ascidian miR-92 family members form a cluster and might play potentially crucial regulators in ascidian development.

### 3.2. The Origin and Evolution of csa-miR-92 in C. savignyi

To investigate the evolutionary relationship of ascidian mir-92, we constructed three phylogenetic trees using the mir-92a, mir-92b, and mir-92c precursor sequences collected from the public database. The phylogenetic tree of mir-92a (Figure 2A) contained two clades; one clade included csa-mir-92a, cro-mir-92a, odi-mir-92a, and sko-mir-92a from Urochordata (*C. savignyi*, *C. robusta*, and *O. dioica*) and Hemichordata (*Saccoglossus kowalevskii*), suggesting that csa-mir-92a has a close relationship with sko-mir-92a. The mir-92b phylogenetic tree (Figure 2B) contained one clade, including csa-mir-92b, cro-mir-92b, odi-mir-92b, cte-mir-92b, and dps-mir-92b, showing that csa-mir-92b has a close relationship with cte-mir-92b. For the mir-92c, two clades were presented in the phylogenetic tree (Figure 2C). Csa-mir-92c was presented with cro-mir-92c, dps-mir-92c, and sko-mir-92c together forming a clade. In summary, the urochordate miR-92 family members (csa-mir-92, cro-mir-92, and odi-mir-92) have a close kinship with *D. pseudoobscura* (dps-mir-92), *S. kowalevskii* (sko-mir-92), and *C. teleta* (cte-mir-92) in terms of origin and evolution.

### 3.3. The mir-17-92/mir-92 Gene Family in Urochordata and Other Model Animals

To further explore the evolution of the miR-17-92 cluster in different species, we compared the genomic location of csa-mir-92 members in 12 species (Figure 3). In vertebrates, the miR-17-92 clusters contained six members, including mir-17, mir-18a, mir-19a, mir-20a, mir-19b, and mir-92a-1. The miR-106a-303 cluster contained six members, including mir-106a, mir-18b, mir-20b, mir-19b-2, mir-92a-2, and mir-363. It was evident that the miR-17-92 and miR-106a-303 clusters were conserved in species with a high evolutionary status, such as *H. sapiens*, *M. musculus*, *G. gallus*, and *D. rerio* [35]. However, in *X. laevis* was a lack of mir-19b in the mir-17-92 cluster, which had five members, including mir-17-2, mir-18b, mir-19a, mir-20a, and mir-92a. We further found that mir-17, mir-18a, mir-19a, mir-20a, and mir-19b were absent in invertebrates. Only the mir-92 family members existed in the invertebrate genome and were grouped into a cluster. For Urochordata, *C. savignyi* had a miR-92 cluster including three members: csa-mir-92a, csa-mir-92b, and csa-mir-92c. *C. robusta* had two miR-92 clusters: one cluster including cro-mir-92a, cro-mir-92c, and cro-mir-92d localized on chromosome 3p: 1864426–1865319 and another cluster including cro-mir-92b and cro-mir-92e localized on chromosome 1p: 2635396-2635776. Together, these results indicated that the mir-17-92 gene cluster was an evolutionary innovation that was uniquely present in the species with a high evolutionary status.

### 3.4. Expression Profiles of the csa-miR-92 in C. savignyi

To understand the expression patterns of the csa-miR-92 family members, we validated the expression profiles of three csa-miR-92 members at different embryonic stages (14 hpf, 18 hpf, 21 hpf, 24 hpf, 28 hpf, 32 hpf, and 42 hpf) and various tissues (gonad, intestines, heart, blood, muscle, and stomach) using qPCR. The results showed that three csa-miR-92 members were expressed in all the examined developmental stages. Their expression levels in 42 hpf larvae dramatically increased (*p* < 0.01) compared with those at the earlier stages (14–32 hpf) (Figure 4A), indicating that csa-miR-92 family members play vital roles in larval tail regression and metamorphosis. Further examinations of the expression profiles in various tissues showed three csa-miR-92 family members that were expressed in all the examined adult tissues and predominantly (*p* < 0.01) in the heart and muscle (Figure 4B).

The spatial expression of csa-miR-92c at different larval stages was examined by in situ hybridization (Figure 5 and Appendix A). The results showed that the expression of csa-miR-92c was distributed throughout the embryo at the early stage (10 hpf) and larval tail regression stage (42 hpf), which is consistent with the results of the qPCR. At the larvae stages 18 hpf, 21 hpf, 24 hpf, and 31 hpf, stronger expression signals of csa-miR-92c were showed in the trunk. They were expressed in the epithelial cells and notochord cells as well (Figure 5). With the elongation of the tail, the positive signals in notochord cells turned to be weak after the lumen connection (Figure 5). These results suggested that csa-miR-92c was involved in the regulation of *Ciona* embryogenesis and larval development, especially in metamorphosis and notochord tubulogenesis.

### 3.5. Knockout (KO) of csa-miR-92c by CRISPR-Cas9 Resulted in Disorganization of the Notochord Cells and the Failure of Lumen Formation

Among three members of miRNA92, csa-miR-92c was expressed at a relatively higher level compared with the other two members (Figure 4A). We therefore picked it up to explore its developmental roles in *Ciona* organogenesis. We designed three gRNAs at three locations of csa-miR-92c, including the mature sequence, the active site of Dicer, and the active site of Drosha, to induce mutation by the CRISPR-Cas9 approach (Figure 6A and Appendix A). After validation of the efficiency of three gRNAs (Appendix A), we co-electroporated the Cs-Brachyury>NLS::Cas9::NLS::P2A::mCherry and the PCR products of three sgRNAs mixture into the fertilized eggs and examined the phenotypes in gRNAs-injected embryos. The results showed that the arrangement of notochord cells in the gRNA-injected groups was disrupted compared with that in the control groups. In 16 hpf-staged embryos, the Cas9-positive notochord cells in csa-miR-92c gRNA-injected embryos became narrow and were squeezed together. The yellow dashed box indicated the disordered notochord cells (Figure 6B). These cells either failed to complete the cell intercalation process or had the problems with the arrangement of notochord cells after the intercalation stage. With this development, the cas9-positive notochord cells were further failures during lumen formation (Figure 6B). The yellow-dashed box showed that the lumen connection and tube formation were uncompleted in cas9-positive embryos compared with the control groups. The number and percentage of the csa-miR-92c-KO embryos were shown in Appendix A. Overall, these results suggest that csa-miR-92c plays an essential role in the arrangement of ascidian notochord and lumen formation and coalescence in notochord tubulogenesis.

### 3.6. Target Prediction of csa-miR-92 and Gene Ontology and KEGG Enrichment Analysis

MiRNAs function through targeting complementary sequences in mRNA molecules to regulate diverse biological processes. To further know about the molecular mechanisms and biological roles of csa-miR-92 family members, we predicted their target genes by using miRanda and targetscan software. A total of 984 putative target genes were identified. Moreover, we screened the target genes by their expression and collected the downregulated genes (Appendix A) from 18 hpf to 42 hpf. GO analysis revealed that the target genes were enriched in the biological processes, molecular functions, and cell components (Figure 7A and Appendix A). The target genes found in the membrane-enclosed lumen or cell junction indicated the possibility that csa-miR-92 modulated the notochord lumen formation via these molecular targets. To explore the signal pathways that csa-miR-92 target genes were involved in, we collected 550 target genes that were annotated with the KEGG database for further analysis. The results showed that these genes were categorized into six parts, including organismal systems, cellular processes, genetic information process, metabolism, human diseases, and the environmental information process (Figure 7B and Appendix A). Furthermore, we found that many important signaling pathways were involved in the putative target genes of csa-miR-92, such as the fibroblast growth factor (FGF), transforming growth factor beta (TGFβ), mitogen-activated protein kinase (MAPK), wingless/integrated (Wnt), nuclear factor kappa-B (NF-κB), nodal growth differentiation factor (Nodal), and planar cell polarity (PCP) signaling pathways (Figure 7C). These pathway components are likely to represent the indirect targets of Brachyury-induced signaling in the notochord.

## 4. Discussion and Conclusions

In this study, we found three miRNAs (csa-miR-92a, csa-miR-92b, and csa-miR-92c) that formed a miRNA cluster in the genome of *C. savignyi* and were highly conserved across species. The miR-92 cluster was predominantly expressed in the notochord cells in *Ciona* embryos during 18-31 hpf. Moreover, the knockout of the csa-miR-92c resulted in the disarrangement of the notochord cells, which might be regulated through the Wnt/PCP or NF-κB signaling pathways. These data together indicated that the miR-92 cluster plays vital roles in *Ciona* notochord development during embryogenesis.

miRNAs are small noncoding RNAs. Many of them are clustered in the genomes of animals, which can be independently or simultaneously transcribed into a single polycistronic transcription [6]. The miR-17-92 cluster is a typical highly conserved polycistronic miRNA gene [36]. In humans, the miR-17/92 cluster together with its two paralogs comprises miRNAs: miR-17, miR-18a, miR-19a, miR-20a, miR-19b-1, miR-92a-1, miR-106b, miR-93, miR-25, miR-106a, miR-18b, miR-20b, miR-19b-2, miR-92a-2, and miR-363 [9]. However, miR-92c or miR-92d were identified in the invertebrate genome but not in vertebrates. We found three miRNAs (csa-miR-92a, csa-miR-92b, and csa-miR-92c) that formed a miRNA cluster in the *Ciona* genome, while other homologs of vertebrate miR-17-92 cluster members could not be identified. Other members, except for miR-92a/b, of the vertebrate miR-17-92 cluster also disappeared in many invertebrates. A previous study showed that, as the species diversity increased during evolutionary history, the varieties of distribution of miRNA genes also increased, catering to the individual and interactive effects of multiple complex evolutionary forces [35]. The low conservation of the miR-17-92 clusters between vertebrates and invertebrates might result from genome duplication in vertebrates.

Since the *Ciona* genome does not have the intact members of miR-17-92 cluster as vertebrates, we explored whether three members of miR-92 cluster identified in *Ciona* could form a cluster and effectively perform similar functions as the vertebrate miR-17-92 clusters do. Our results showed that the csa-miR-92 was ubiquitously expressed in different tissues and at different stages of embryo, identical to the miR-17-92-cluster, which was highly expressed from the embryonic cells to adulthood [37]. Noteworthy, three *Ciona* miR-92 family members had consistent expression levels in adult tissues or at different embryonic stages, indicating that they either were regulated by the same upstream signals or processed similar stability. In a word, the *Ciona* miR-92 cluster members identical conserved seed sequences and similar expression levels. Thus, we speculated that three miR-92 cluster members cooperated in the regulation of the same signaling pathways so that their functions were more effective in controlling orderly biological activity. However, it is worth noting that the six miRNA members in the vertebrate miR-17-92 cluster were often expressed at different levels. The transcriptional level of miR-92a was the most highly expressed members compared to that of the other miRNA members [38,39]. The cooperation of the miR-17-92 cluster members occurred in modulating a specific pathway, such as TGF-β signaling [40,41]. Thus miR-17-92 cluster had overlapping or complementary functions. We identified 984 putative target genes of csa-miR-92 in this study, which involved multiple biological processes. By comparison, between predicted target genes of csa-miR-92 family and reported targets of the mammalian miR-92 cluster, we found that the seed sequence of csa-miR-92 was conserved, but its target genes were diverse from vertebrates. However, csa-miR-92s might sometimes maintain their regulation of a conserved signal pathway by targeting different genes, even if they lost the interaction with a conserved target in vertebrates, which suggested that conservation of miRNA mediated transcriptional regulation system might exceed sequence level and live up to progress level.

*Ciona* notochord is made of 40 cells, arranged in a single row in the midline of the larval tail. In the early development of notochord, cells intercalate into a column during convergent extension. Then, extracellular pocket lumen forms, expands, and merges into a single lumen within the notochord [30,42,43]. Here, we observed that csa-miR-92c was expressed in the notochord cells at different embryonic stages. Furthermore, the knockout of csa-miR-92c by CRISPR/Cas9 resulted in a disarrangement of the notochord cells and failure of lumen formation. To detect the molecular mechanism of this phenotype, we screened the target genes of miR-92. Notably, some molecular networks underlying notochord intercalation and extension movements were identified as the target candidates, including FGF [44,45,46], Wnt/PCP [47], and NF-κB pathways and solute carrier family 26 (SLC26) [48]. These results suggested that the csa-miR-92 cluster was involved in the regulation of notochord development via multiple pathways. In general, our study revealed the phylogenesis, expression, and the roles of the miR-92 cluster in *Ciona* organogenesis. The underlying mechanisms and various additional roles need to be further explored in the future.

## Figures and Tables

**Figure 1 genes-12-00406-f001:**
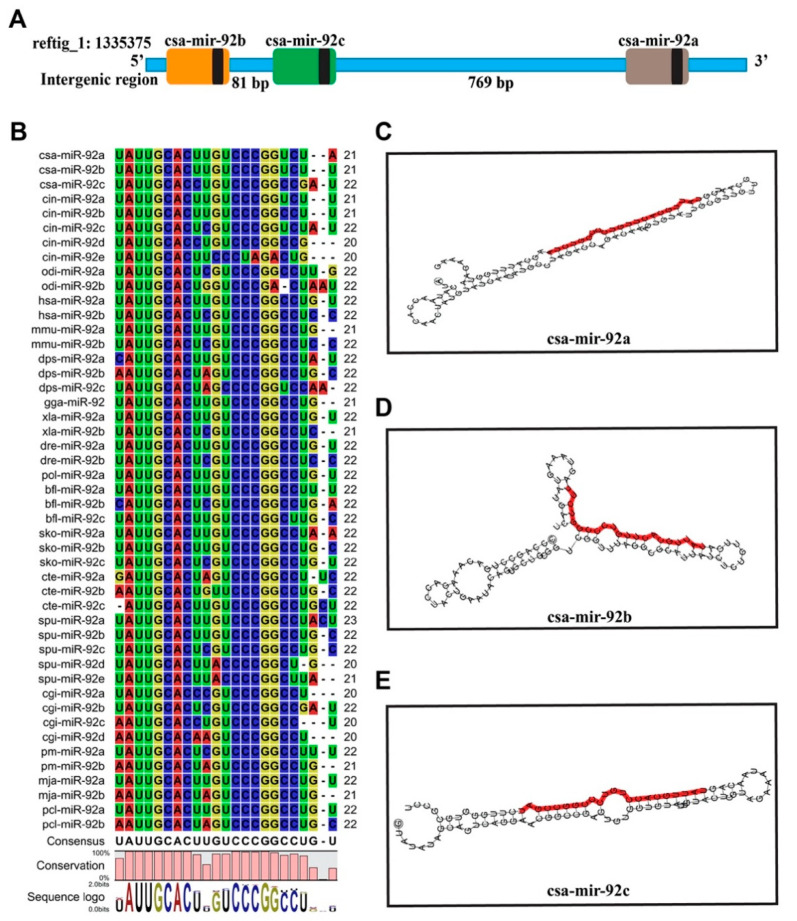
Distribution and second structures of miR-92 members in *Ciona savignyi*. (**A**) Distribution of miR-92 members in the *C. savignyi* genome. The blue line indicates the partial sequence (5′-3′) of the genome; the orange, green, and gray square represents the pre-microRNA (miRNA) of csa-mir-92b, csa-mir-92c, and csa-mir-92a, respectively. The black square represents the mature miRNAs. (**B**) The multiple sequence alignment of different species of the mature miR-92 sequence. Each row represents a miRNA. Red, green, yellow, and blue color highlight “A“, “U“, “G“, and “C“, respectively. “AUUGCA“ represents the seed sequence of miR-92. “csa“, “cin“, “odi“, “hsa“, “mmu“, “dps“, “gga“, “xla“, “dre“, “pol“, “bfl“, “sko“, “cte“, “spu“, “cgi“, “pm“, “mja“, and “pcl“ represent *C. savignyi*, *C. robusta* (former name: *C. intestinalis* type A), *Oikopleura dioica*, *Homo sapiens*, *Mus musculus*, *Drosophila pseudoobscura*, *Gallus gallus*, *Xenopus laevis*, *Danio rerio*, *Paralichthys olivaceus*, *Branchiostoma floridae*, *Saccoglossus kowalevskii*, *Capitella teleta*, *Strongylocentrotus purpuratus*, *Crassostrea gigas*, *Pinctada martensii*, *Marsupenaeus japonicas*, and *Procambarus clarkia*, respectively. (**C**–**E**) The secondary structures of csa-mir-92a, csa-mir-92b, and csa-mir-92c. The red colorful base pairs represent the mature sequences of the miRNAs.

**Figure 2 genes-12-00406-f002:**
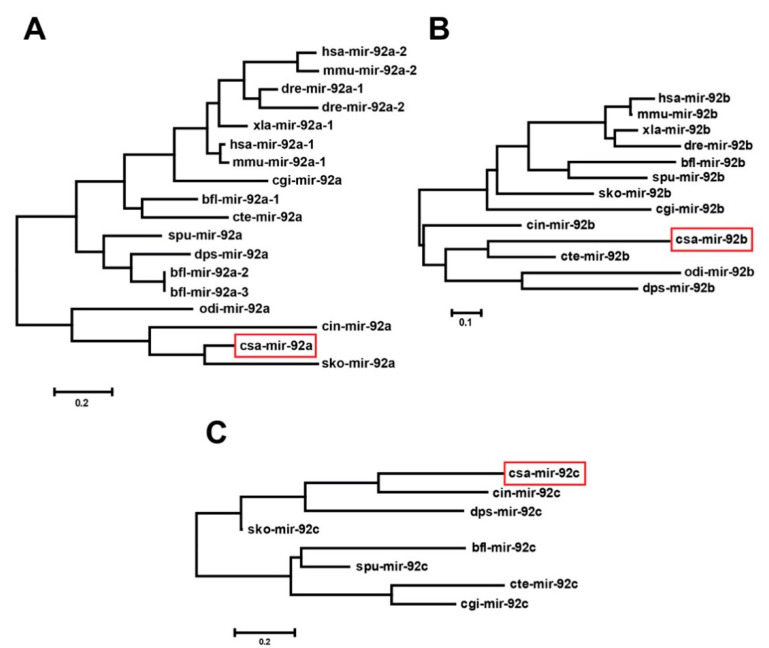
The phylogenetic trees of miR-92 in different species. (**A**–**C**) Phylogenetic trees of the miR-92a (**A**), miR-92b (**B**), and miR-92c (**C**) precursor sequence by the Maximum Likelihood method. The red rectangles indicate csa-mir-92a, csa-mir-92b, and csa-mir-92c, respectively. Scale bar = 0.2 in (**A**,**C**) and 0.1 in (**B**).

**Figure 3 genes-12-00406-f003:**
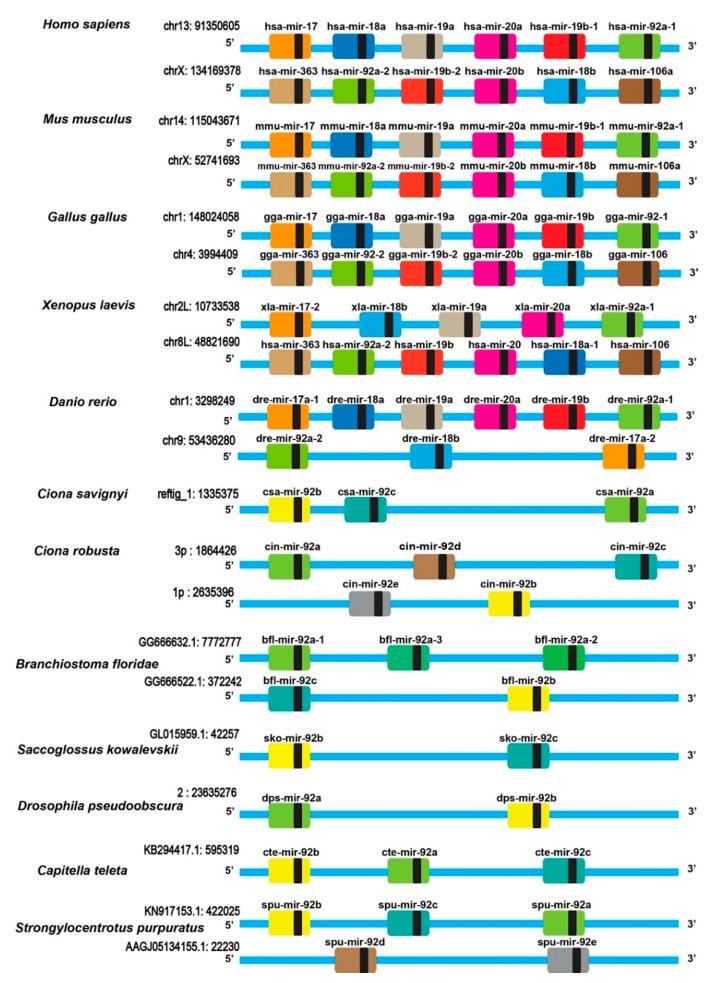
The schematic diagram of the mir-17-92/mir-92 cluster in different species. Each larger-colored square represented a different miRNA. The smaller-black squares characterized the mature miRNAs. The blue line indicated the partial genome. The genomic location of the mir-17-92/mir-92 cluster gene was labeled on the left-top of the schematic diagram. The left rows are the name of the species.

**Figure 4 genes-12-00406-f004:**
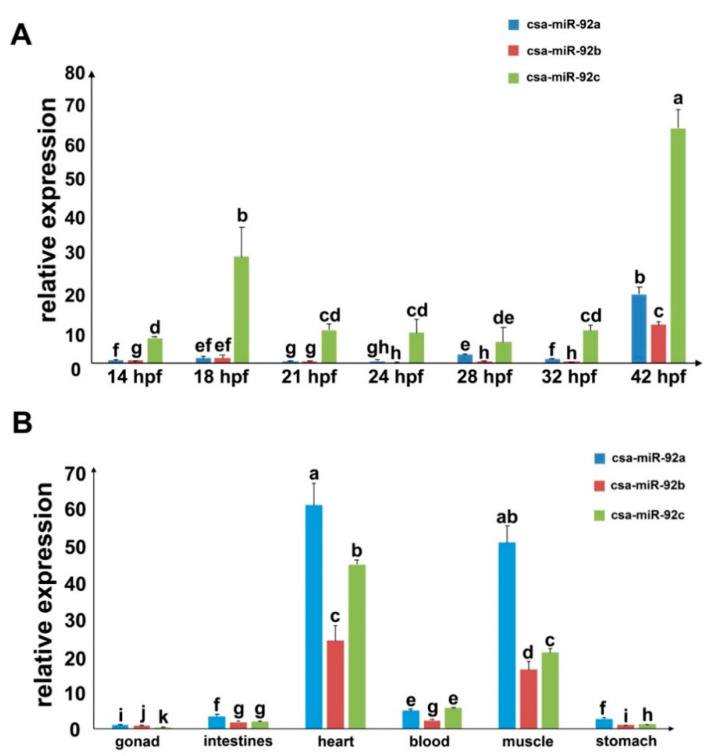
Expression profiles of csa-miR-92 miRNAs. (**A**) Relative expression of csa-miR-92a, csa-miR-92b, and csa-miR-92c at different stages (14 hours postfertilization (hpf), 18 hpf, 21 hpf, 24 hpf, 28 hpf, 32 hpf, and 42 hpf). (**B**) Relative expression of csa-miR-92a, csa-miR-92b, and csa-miR-92c in various tissues (gonad, intestines, heart, blood, muscle, and stomach). The relative expression of each miRNA was calculated using the 2^−ΔΔCt^ method and normalized to the U6 gene expression. Results represented the mean of triplicate assays with the standard error (mean ± SD). Significance (*p* < 0.05) was indicated by a, b, c, d, e, f, g, h, i, j, and k.

**Figure 5 genes-12-00406-f005:**
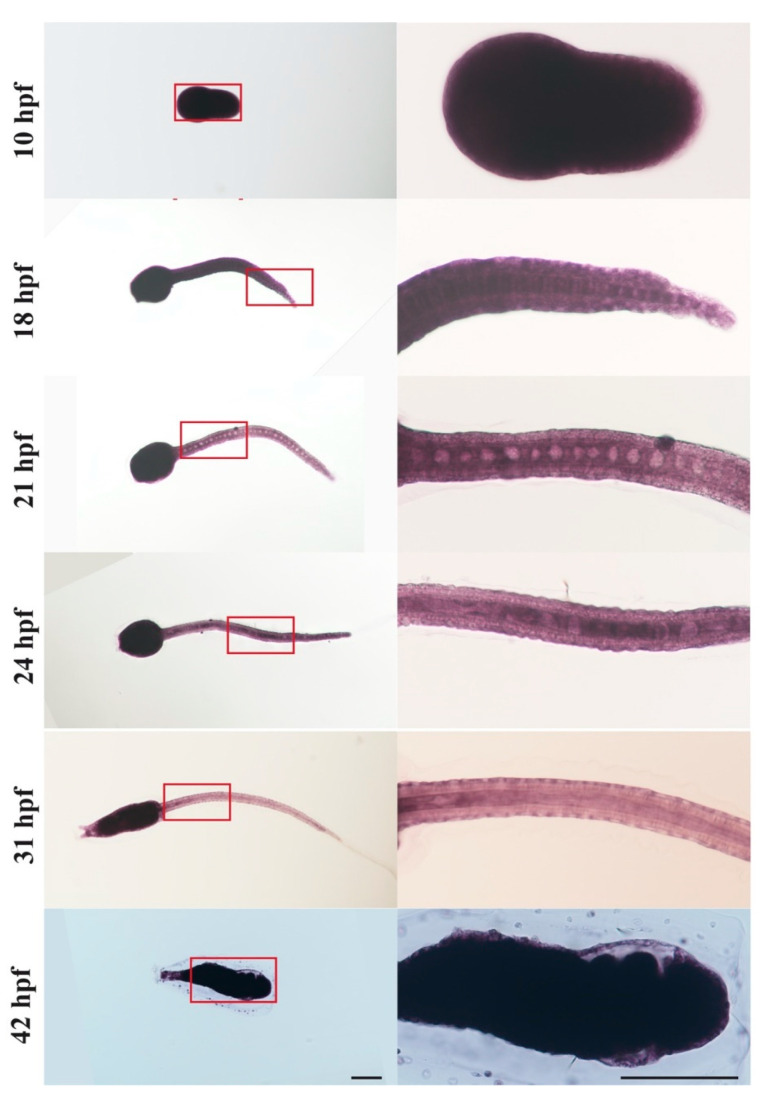
Expression patterns of csa-miR-92c detected by in situ hybridization. Embryos and larvae at 10, 18, 21, 24, 31, and 42 hpf were hybridized with locked nucleic acid (LNA) probes of csa-miR-92c. The developmental stages were indicated. The signals of csa-miR-92c were detected in the whole body at 10 hpf and 42 hpf and expressed in the trunk, notochord cells, and epithelial cells at 18, 21, 24, and 31 hpf. The red frame indicated the regions of zoom-in images in the middle column. Scale bars represent 100 μm.

**Figure 6 genes-12-00406-f006:**
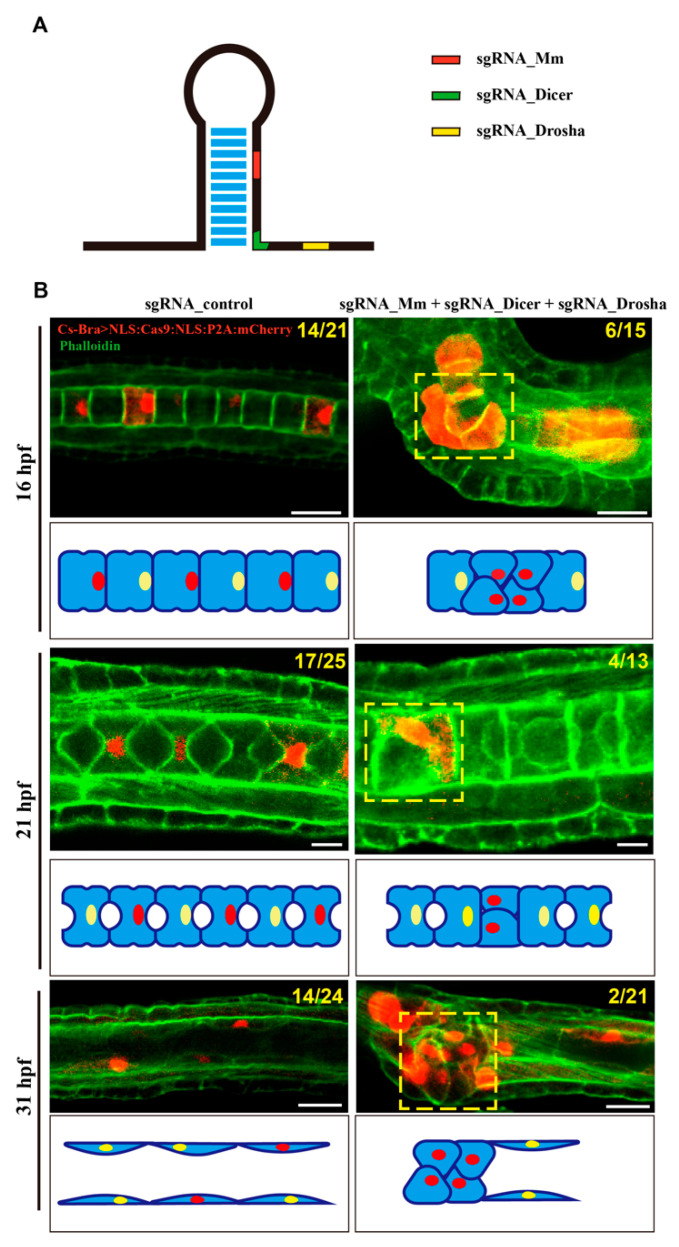
The phenotypes of csa-miR-92c-KO embryos in *Ciona savignyi*. (**A**) The schematic diagram of csa-miR-92c single guide RNA (sgRNA) in *C. savignyi*. The red, green, and yellow areas represented the sgRNA in the mature sequence, the Dicer site, and the Drosha site of csa-miR-92c, respectively. The blue area indicated the hydrogen bond. (**B**) Confocal images of csa-miR-92c KO embryos (right) and control ones (left) at 16 hpf, 21 hpf, and 31 hpf stages, respectively. The PCR product of sgRNA (50 μL), the plasmid of Cs-Brachyury(3k)>NLS::Cas9::NLS::P2A::mCherry (30 μg) and electroporation buffer (420 μL) were electroporated into fertilized eggs (300 μL). After electroporation, embryos were allowed to develop at 18 °C to 16 hpf, 21 hpf, and 31 hpf for confocal observation. The ratio of embryos with phenotypes and cas9-positive embryos were labeled on the top right of the images in the knockout (KO) groups. The ratio of normal embryos and cas9-positive embryos were labeled on the top right of the images in the controls. The notochord cells, which were forced to express cs-brachyury>NLS:Cas9:NLS:P2A:mCherry, were shown in red, while phalloidin labeled the cytoskeleton (green). The yellow-dashed box indicated the abnormal arrangement of the notochord cells. The schematic images below the confocal images described the phenotypes of the csa-miR-92c KO and control. The blue, yellow, and red areas represented the notochord cells, the nuclei, and the nuclei in cells expressed in cs-brachyury>NLS:Cas9:NLS:P2A:mCherry (red), respectively. Scale bars represent 10 μm.

**Figure 7 genes-12-00406-f007:**
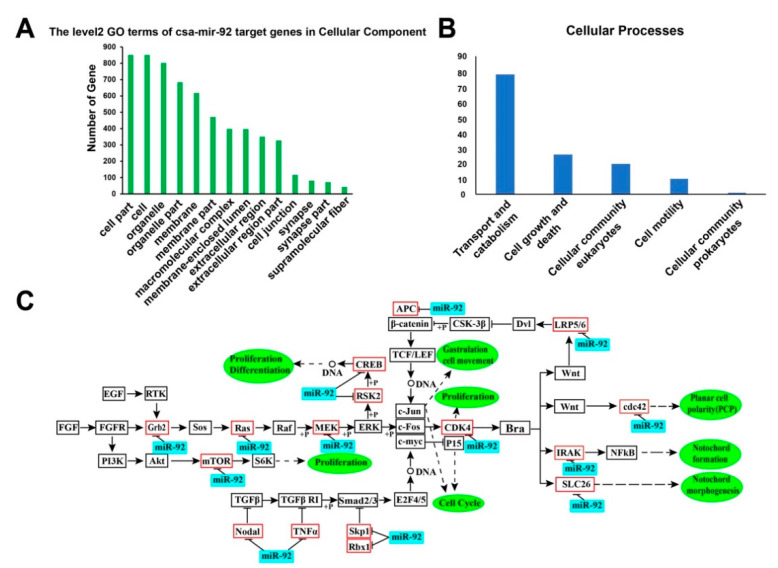
The Gene Ontology (GO) and Kyoto Encyclopedia of Genes and Genomes (KEGG) enrichment analysis of the target genes and signaling pathways of the related target genes of miR-92 in *Ciona*. (**A**) Partial view of the level2 GO terms of the csa-miR-92 target genes in the cellular component of Gene Ontology. The abscissa represents the classification of the ontologies of GO, and the ordinate represents the number of genes contained in specific category entries. (**B**) The cellular processes in the KEGG pathway annotation. The abscissa described five cellular processes, including transport and catabolism, cell growth and death, cellular community eukaryotes, cell motility, and cellular community prokaryotes. The ordinate indicated the number of genes in the cellular processes. (**C**) The signaling pathways related to the putative target genes of csa-miR-92, including fibroblast growth factor (FGF), transforming growth factor beta (TGFβ), mitogen-activated protein kinase (MAPK), wingless/integrated (Wnt), nuclear factor kappa-B (NF-κB), nodal growth differentiation factor (Nodal), and planar cell polarity (PCP) signaling. Red rectangles represented the predicted target genes of csa-miR-92; black rectangles represented the genes in a pathway. Csa-miR-92 was shaded by rectangles filled with blue color; black arrows, black line bar, and dashed arrow denoted positive regulation, negative regulation, and the biological regulation process, respectively.

## Data Availability

Not applicable.

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
