# Peer review of "MiR-92 Family Members Form a Cluster Required for Notochord Tubulogenesis in Urochordate Ciona savignyi"

_genes, 2021, doi:10.3390/genes12030406_

Round 1
Reviewer 1 Report
The manuscript by Yang et al. reported characterization of miR-92 family in C. savignyi. The authors investigated developmental expression profile and tissue distribution of csa-miR-92s. They also showed csa-miR-92c expression in the notochord, and its involvement of notochord formation by knockdown experiment. Overall, this study provides interesting insights into micro-RNA function in urochordate. I hope authors find my comments helpful to improve the manuscript.
Major comments
- This study showed roles of miR-92 clusters in notochord development of ascidian. Is it the case in vertebrates or other invertebrates? Functional roles of micro RNAs are summarized in the Introduction. But their roles in notochord development in other animal species are not well described. Please clarify.
- Related to comment 1. The authors mentioned that “Our results provide insights on the understanding of miR-92 ….during evolution.” (Page 1, line 31-32). This sounds like overstatement for me. To support this, authors should show examples that miR-92 functions are conserved between Ciona and other organisms. For instance, the authors introduced about miR-92 function in cardiomynocyte. Is this the case in savignyi ? Fig. 4B represents csa-miR-92 expression in heart and muscle. Also, Fig. 5 shows ubiquitous staining of csa-miR-92c in the trunk, which may include trunk ventral cells. Are there any defects in heart formation?
- Section 3.5. CRISPR/Cas9 for csa-miR-92s sounds interesting. But, I ask authors to describe more details. First, authors should include Figure or Table for the mutated nucleotide sequences induced by the three gRNAs. Second, what the “validation of the efficiency” (Page 9, Line 269) means? The authors used Brachury > Cas9 construct for notochord-specific KO. Did the authors tested whole-body effects using other construct or injection of Cas9 mRNA/protein? If so, I suggest authors to include supplemental Figure about this. Finally, how many embryos are tested, and how much percentage of electroporated embryos developed normally? Please clarify.
Minor comments
- Page 4, line 144-148. Please move them to the Introduction. They are background information and rationale.
- Page 9, line 266. its role > its functional (or developmental) role
- What are putative targets genes of the csa-miR-92s? I suggest the authors to mention this in the Discussion.
Reviewer 2 Report
In this article the authors identified three miRNAs (csa-miR-92a, csa-miR-92b, and csa-miR-92c) that belong to the miR-92 family and form a miRNA cluster in the genome of a urochordate marine ascidian Ciona savignyi. They found that the mature sequences of urochordate miR-92 family members were highly conserved compared with vertebrate species.
Although the evolution analysis of this miRNAs results very interesting, the characterization of their expression and their function is insufficient and incomplete to support the author's conclusions. In particular:
- which is the time point of the expression analysis by qRT-PCR in the different tissues (Fig. 4B)? The miR-92c seems the most expressed in the fig. 4A at 42hpf, but it is not the highest in heart and muscles, where these miRNAs seems prevalently expressed (Fig. 4B). Although they are in the same cluster and are transcribed as polycistronic transcript, could be possible that they present different regulation in the maturation process that could spacially regulate them differently in the different tissue?
- which are the possible promoter regions and the possible transcription factor that activate their expression?
- Why in the In situ hybridization experiment the authors analysed only miR-92c?
- How they validate the efficiency of the three gRNAs? Did they analyse the genome sequence? did they analyse the possible OFF targets in the Ciona genome?
- what is the "control-injected group"? Are the eggs injected with a mix of scramble gRNAs?
- After electroporation in fertilized eggs how they demonstrate that the csa-miR-92c is effectively knock-out? did they analyse the genome sequence? did they analyse the OFF targets? did they measure the level of mature miR-92c? How can they be sure that the phenotype is related to miR-92c loss of function and not an OFF targets effect?
- how they validate the specificity of the gRNAs for miR-92c? since the three miRNAs of the cluster present high similarity in sequence, how are they sure that they are not affecting also the other two miRNAs? especially for the use of the gRNA that targets the mature sequence of the miRNA. It has been reported that if in similar sequences there are mismatches equal to or less than two, CRISPR/cas9 system can unbiasedly cleave the other target sequences (DOI: 10.1038/srep22312). Did they measure the levels of the mature form of the other members of the cluster?
- How many eggs they injected? how many animals they analyse? which is the % of animals that show the phenotype? Is this phenotype statistical significant?
- In the final parte of the work, the authors identified possible pathways regulated by the miR-92a/b/c in Ciona using bioinformatic prediction. The authors should validate the alteration of these pathways in vivo upon the miR-92c KO. At least they should show by qRT-PCR the alteration at transcript levels of those genes reported as possible direct targets of these miRNAs in fig. 7C, and possibly validate them as real bound targets by using luciferase assay.
- Finally, if the authors want to link the miR-92c mediated regulation of this pathway to the miR-92cKO phenotype, they should try (by using drugs or shRNA) to restore the transcript/protein levels of that targets and rescue the phenotype
Round 2
Reviewer 2 Report
The authors improve some aspect of the data presentation but there are still some points that could be ameliorated:
- specified also in materials and methods that the qRT-PCR on the different tissues is on adult tissues (line 102)
- in material and methods section they should describe the T7 test, showed now in figure S2. It is not reported on which cellular material has been performed the test: total embryo? at which stage? after how long after electroporation? too many details are missed
- in their response to point 4 the authors stated that they "didn’t analyze the off-target experimentally." In point 6 they stated "Yes, we did analyze the genome sequence and the possible OFF targets in the Ciona genome." I'm confused by their answers, but I think that in silico predictions are not sufficient to ensure the good work of the gRNAs. They should test experimentally the not-activities of the used gRNAs on the predicted OFF target sequences, as well as on the others miRNA sequences of the cluster. This could indicate that the gRNAs are specific for the miR-92c. Otherwise it is impossible to determine that the observed phenotype is due to the effect of miR-29c loss of function. If you already have or produce this data, please provide in the figures the appriopiate controls of the used technique
- The sentence "The phenotype was determined through observations from large amount animals and comparison with the controls." in response to point 6, does not make any scientific sense. As in response to point 8, please provide in the material and methods section, and in the figure legend of figure 6, the total number of electroporated embryos and at which time point of development the embryos have been electroporated. For each time point of analysis how many embryos have been analysed and how many of that present the phenotype at each time point of the analysis.
- in their response to point 8 the authors stated that they electroporated "several hundred of eggs" (very generic number), and that they "examined 15, 13, 21 cas-postive embryos" for each time point. How much is the total number of cas-positive embryos on the "hundreds" injected? In other words, 15,13,21 of examined embryos which % are respect to the total cas-9 positive embryos.
- The percentage of the embryos that showed phenotype was around 10-40%. It is a big range. It means that on 20 embryos analysed (the biggest group analysed for a specific time point) between 2 and 8 showed the phenotype. Considering that only 15,13 and 21 embryos are analysed on hundred electroporated embryos, it seems that the total number of embryos that show the phenotype is very low. Please provide these numbers, and the relative percentage, in a table.
- The authors added the qRT-PCR for some predicted target at different time point of development. This is a very interesting data but the data presentation presents still defects in the information provided. It is unclear how they selected this set of predicted transcripts for the analysis. Please, in the material and methods section, indicate which is the reference used, how many embryos were used for each time points, how many biological replicates have been done. Please add the error bars to graph and the statistics that could indicate that these variation between time points are significant
